# Novel Matrine Derivatives as Potential Larvicidal Agents against *Aedes albopictus*: Synthesis, Biological Evaluation, and Mechanistic Analysis

**DOI:** 10.3390/molecules28073035

**Published:** 2023-03-29

**Authors:** Song Ang, Jinfeng Liang, Wende Zheng, Zhen Zhang, Jinxuan Li, Zhenping Yan, Wing-Leung Wong, Kun Zhang, Min Chen, Panpan Wu

**Affiliations:** 1School of Biotechnology and Health Sciences, Wuyi University, Jiangmen 529020, China; 2International Healthcare Innovation Institute (Jiangmen), Jiangmen 529040, China; 3The State Key Laboratory of Chemical Biology and Drug Discovery, Department of Applied Biology and Chemical Technology, The Hong Kong Polytechnic University, Hung Hom, Kowloon, Hong Kong SAR, China

**Keywords:** matrine derivatives, synthesis, anti-mosquito activity, mechanistic analysis

## Abstract

A large number of studies have shown that matrine (**MA**) possesses various pharmacological activities and is one of the few natural, plant-derived pesticides with the highest prospects for promotion and application. Fifty-eight **MA** derivatives were prepared, including 10 intermediates and 48 target compounds in 3 series, to develop novel mosquitocidal agents. Compounds **4b**, **4e**, **4f**, **4m**, **4n**, **6e**, **6k**, **6m**, and **6o** showed good larvicidal activity against *Aedes albopictus*, which is both a highly aggressive mosquito and an important viral vector that can transmit a wide range of pathogens. Dipping methods and a bottle bioassay were used for insecticidal activity evaluation. The LC_50_ values of **4e**, **4m**, and **6m** reached 147.65, 140.08, and 205.79 μg/mL, respectively, whereas the LC_50_ value of **MA** was 659.34 μg/mL. Structure–activity relationship analysis demonstrated that larvicidal activity could be improved by the unsaturated heterocyclic groups introduced into the carboxyl group after opening the D ring. The **MA** derivatives with oxidized N-1 lost their mosquitocidal activities, indicating that the bareness of N-1 is crucial to maintain their anti-mosquito activity. However, the activity was not greatly influenced by introducing a cyan group at C-6 or a benzene sulfonyl group at N-16. Additionally, compounds **4e** and **4m** exhibited good inhibitory activities against acetylcholinesterase with inhibitory rates of 59.12% and 54.30%, respectively, at a concentration of 250 μg/mL, whereas the inhibitory rate of **MA** was 9.88%. Therefore, the structural modification and mosquitocidal activity of **MA** and its derivatives obtained here pave the way for those seeking strong mosquitocidal agents of plant origin.

## 1. Introduction

Matrine (**MA**), a quinolizidine alkaloid and an important natural product, is isolated from the plant species of the family Fabaceae, Sophora flavescens Aiton, Sophora tonkinensis Gagnep, *Sophora alopecuroides* L. [1,2,3]. **MA** and its analogs possess a variety of biological properties, such as anticancer activity, anti-inflammatory activity, insecticidal activity, antimicrobial activity, and antiviral activity; **MA** alkaloids are excellent precursors for structural modification and thus have attracted great interest from scholars [4,5,6,7,8,9,10,11,12,13,14,15]. As an insecticide, **MA** has remarkable insecticidal activity against a variety of agricultural pests, such as *Bradysia odoriphaga* Yang et Zhang, *Cnaphalocroci smedinalis* (Guenee, 1854), *Ectropis obliqua hypulina* Wehrli, *Clostera anachoreta* Denis and Schiffermüller,1775, *Eriosoma lanigerum* (Hausmann), *Psylla chinensis* Yang et Li, *Mesonura rufonota* Rohwer, and *Aceri macrodonis* Keifer, with contact toxicity and gastric toxicity as the main modes of action [16,17,18,19,20]. As a result of the wide range of biological activities against insects, we are interested in studying the anti-mosquito activity of **MA** and its derivatives against *Aedes albopictus* (Skusse).

*Ae. albopictus*, an insect of the mosquito family Cullicidae and genus Aedes, is one of the most important vectors for the transmission of nearly 80 types of viral diseases, including yellow fever, Venezuelan equine encephalitis, Chikungunya, West Nile virus disease, and Ross River fever [21,22,23,24,25]. Currently, chemical, biological, physical, and other methods, such as maintaining personal and environmental hygiene, are applied to reduce the mosquito population, and chemical insecticides are generally accepted as the most effective tactic for controlling mosquitoes [26,27,28]. However, the negative effects of chemical insecticides, such as potential health risks, water contamination, environmental pollution, and toxicity to nontarget organisms have increased considerably [29,30,31,32]. Therefore, with the development of science, plant-derived mosquito insecticides, which belong to a large class of green pesticides, have become one of the mainstays of pesticide development [33,34,35].

**MA** has become one of the few plant-derived pesticides with the greatest promotion and application prospects owing to its good insecticidal, antibacterial, growth-regulating, and other biological activities, which make them potential candidates as mosquitocidal agents [36,37,38]. In the present study, we designed, synthesized, and characterized three series of **MA** derivatives by structural modification, and screened these derivatives for their potential larvicidal and adulticidal activity against the mosquito vector, *Ae. albopictus*. The structure–activity relationships (SARs) of **MA** and its derivatives with anti-mosquito activity were obtained. Furthermore, the effect of the more active compounds on the larval growth cycle was investigated, and the anti-mosquito mechanism was explored. Therefore, our study on the structural modification and anti-mosquito activities of **MA** and its derivatives provides guidance to further accelerate research and development of **MA** as a plant-derived anti-mosquito agent.

## 2. Results and Discussion

### 2.1. Chemistry

As shown in Figure 1, two *N*-phenylsulfonylmatrinic methyl esters (**2a** and **2b**) were obtained through the reaction of **MA** with 6 N hydrochloric acid, followed by methanol and phenyl sulfonyl chloride under potassium hydroxide [39]. The hydrolysis of **2a** and **2b** in the presence of sodium hydroxide and methanol produced *N*-phenylsulfonylmatrinic acids (**3a** and **3b**) [40], which were further reacted with different heterocyclic amines to produce different matrinic amides (**4a**–**4p**) [41]. Further, **2a** and **2b** were oxidized with *m*-chloroperoxybenzoic acid (*m*-CPBA) to produce **2c** and **2d**, respectively [15]. Then, the hydrolysis of **2c** and **2d** under the same conditions that produced **3a** and **3b** yielded **3c** and **3d**, respectively. Similarly, **3c** and **3d** were reacted with different heterocyclic amines to afford different matrinic amides (**5a**–**5p**), which were further reacted with trifluoroacetic anhydride (TFAA), followed by trimethylsilyl cyanide (TMSCN), to obtain the target cyan-substituted matrinic compounds (**6a**–**6p**) [42]. Their structures were well characterized by proton nuclear magnetic resonance (^1^H NMR), carbon nuclear magnetic resonance (^13^C NMR), high-resolution mass spectrometry (HRMS), and melting point analysis (see Appendix A).

### 2.2. Biological Evaluation

#### 2.2.1. Insecticidal Activities

**MA** was considered as a promising natural product with various pharmacological activities [43] and the **MA** showed good insecticidal activity [11]. Therefore, structural modification and insecticidal activities were studied to find anti-mosquito agents in this work. The larvicidal activities and structures of **MA** and its derivatives against the 4th instar larvae of *Ae. albopictus* are shown in Table 1, which revealed that the mortalities of the compounds at a concentration of 500 μg/mL ranged from 0% to 100% and suggested that the larvicidal activities could vary substantially with the structural modifications. The result indicated that compounds **4b**, **4e**, **4f**, **4m**, **4n**, **6e**, **6k**, **6m**, and **6o** exhibited good larvicidal activities with mortalities ranging from 50% to 100%, which were much higher than that of the parent **MA** (23.33%). Additionally, the intermediates did not show larvicidal activities with low or no mortalities. Unfortunately, the larvicidal activities of the compounds of series **5** almost vanished. According to the result, changes in the mortalities of the derivatives have no rules when the *para*-position of benzenesulfonyl was replaced by chlorine or bromine. Although, we have obtained **MA** derivatives with better activity than the parent compound, there was still a certain distance when compared with commercially available anti-mosquito agents. However, such structural modification will have certain guiding significance for subsequent studies.

**MA** and several of its derivatives were selected for preliminary activity tests against female *Ae. albopictus*. Unfortunately, the result indicated that the **MA** derivatives had low activities against adult mosquitoes. Larvicidal activity was tested using the microporous plate method, in which compounds were dissolved in water and entered the larva directly through feeding. In comparison, compounds were applied topically to evaluate insecticidal activity against adult mosquitoes, in which compounds were required to infiltrate the epidermis to enter adult mosquitoes and cause disability or death. Chemical toxicity, premedication methods, and study subjects were all recognized to have an impact on the outcomes of insecticidal action. The **MA** derivatives may have an insecticidal activity that is biased toward the larvae rather than the adults because of their strong polarity and hydrophilicity.

#### 2.2.2. Dose–Response Curves on *Ae. albopictus* Larvae

The dose–response curves of **MA** and seven chosen derivatives (**4b**, **4e**, **4m**, **6e**, **6j**, **6g**, and **6m**) were established using the increased concentration test range of the compounds from the results of preliminary activity testing on *Ae. albopictus* larvae as shown in Figure 1. The results showed a dose-dependent pattern for the insecticidal efficacy on *Ae. albopictus* larvae.

The LC_20_, LC_50_, and LC_90_ values for *Ae. albopictus* larvae were determined using the toxicity regression equations that were produced using the dose–response curves, as shown in Table 2. In summary, the LC_50_ values of **MA**, **4b**, **6e**, **6j**, and **6g** were 659.34, 563.90, 436.73, 547.91, and 535.37 μg/mL, respectively. In comparison, **4e**, **4m**, and **6m** showed lower LC_50_ values of 147.65, 140.08, and 205.79 μg/mL, respectively, which indicated that they had a high death rate for larvae. The results showed that compounds **4e**, **4m**, and **6m** had outstanding larvicidal activities and that the LC_50_ value of **MA** was 4.47, 4.71, and 3.20 times the LC_50_ values of these compounds, respectively. The results indicated that the derivatives modified with **MA** did not show good anti-mosquito activity against *Ae. albopictus* when compared to compounds or essential oils that have been reported [44,45]. However, the derivatives showed much higher activity than the parent compounds, which indicated that structural modification of **MA** was beneficial to improve anti-mosquito activity. On the other hand, this study can also provide a preliminary basis for further research on the anti-mosquito activities of **MA** and its derivatives.

#### 2.2.3. Effects of **MA** and Its Derivatives on the Partial Life Cycle of *Ae. albopictus*

The life cycle of *Ae. albopictus* includes eggs, larvae, pupae, and adults [46]. The studies on emergence of surviving larvae treated with drugs and the fecundity of adult female *Ae. albopictus*, which came from the surviving larvae, can prove whether **MA** and its derivatives have an effect on the growth cycle of mosquitoes and provide directions for subsequent studies [47].

##### Effects on the Emergence of *Ae. albopictus* Larvae

As shown in Figure 2, the effects of **MA** and its derivatives (**4e** and **4m**) on larval emergence were tested and compared with the negative control group (dimethylsulfoxide). The eclosion of larvae in the negative control group started on the 3rd day with a rate of 10%. In comparison, larval eclosion started on the 4th day with a rate of 10% for **MA**, the 5th day with a rate of 13% for **4e**, and the 3rd day with a rate of 5% for **4m**. The results indicated that **MA** and its derivatives inhibited eclosion by delaying toxicity and inhibiting larval pupation. The selected **MA** derivatives delayed the emergence time and reduced the emergence rate of *Ae. albopictus* larvae. Additionally, the mortalities of the compound treatment groups increased until the 15th day, in which the mortality rates of the **MA**, **4e**, and **4m** treatment groups were 48%, 29%, and 48%, respectively. In contrast, no death was recorded in the negative control group. The mortalities suggested that the chronic toxicity of **MA** and its derivatives would cause the larvae to fail to transform into pupae and emerge successfully. Therefore, the result of the emergence experiment indicated that maintaining mosquito control is possible by delaying the emergence time and reducing the emergence rate of larvae. Furthermore, this study showed that the derivatives from **MA** had better inhibition on the emergence of *Ae. albopictus* larvae than the parent compound, which was of guiding significance to enhance the larvicidal activity of the parent compound by structural modification of its specific location.

##### Effects on the Fecundity of Adult Female *Ae. albopictus*

There are a variety of techniques for reducing mosquito population density, including killing insects directly with chemicals or equipment or obstructing a particular process of mosquito growth and development [48]. Here, we examined the effects on the fecundity of adult female *Ae. albopictus* that survived from the drug-treated larvae. The average number of eggs laid by adult mosquitoes that emerged from larvae treated with **MA** and its derivatives (**4e** and **4m**) were recorded to explore the effects of **MA** and its derivatives on the fecundity of *Ae. albopictus*. The results are shown in Figure 3. Compared with the control group (dimethylsulfoxide), **MA**, **4e**, and **4m** inhibited the oviposition of the treated female mosquitoes at different concentrations (LC_10_, LC_20_, LC_30_, LC_40_, and LC_50_), indicating that the compounds exhibited a clear effect on the fecundity of adult female *Ae. albopictus*. Furthermore, the average egg-laying rate of the treated female mosquitoes was decreased remarkably by these three compounds as the concentration of the compounds increased, indicating a dose-dependent relationship between the compounds and the oviposition of *Ae. albopictus*. Our findings were consistent with reports that oral feeding of a sublethal concentrations of boric acid reduced the fecundity of females of *Ae. albopictus* [49]. Although the mechanisms underlying these relationships are unknown, one possible explanation is that the drug had an impact on the larvae that lasted until they matured, reducing the quantity of eggs they deposited.

### 2.3. Structure–Activity Relationships

It was reported that **MA** derivatives were obtained by structural modification of **MA**, and their acaricidal activity was six times stronger than that of the parent matrine [15]. In addition, Zhang, et al. synthesized 85 **MA** derivatives: their insecticidal activity against *Oriental armyworm* was tested, and the structure-activity relationship was summarized [50]. In this study, based on the previous synthesis of a large number of **MA** derivatives, a high-throughput screening method was used to determine the anti-larvicidal activity of the compounds against *Ae. albopictus*, and the potential **MA** derivatives were screened. The results of the SAR analysis of these novel **MA** derivatives are summarized in Figure 4. First, larvicidal activity was unacted by the substitution of different halogen atoms (Cl or Br) in R_1_. Second, the compound with hydroxyl as the R_2_ showed low activity against *Ae. albopictus* larvae, which suggested that R_2_ was an important modification site for the optimization of larvicidal activity. Furthermore, the derivatives showed low anti-mosquito activity when the R_2_ was a saturated naphthene or saturated heterocyclic group with nitrogen and oxygen. However, larvicidal activity increased remarkably when the R_2_ was composed of unsaturated heterocyclic groups containing nitrogen or oxygen. Third, no obvious SAR was observed when the R_3_ was a hydrogen atom or a CN group because some derivatives showed a little bit of larvicidal activity after hydrogen was replaced by CN at R_3_, whereas the larvicidal activity of some compounds decreased. Finally, the larvicidal activity of the compounds was almost completely lost when the nitrogen at the N-1 position of **MA** and its derivatives was oxidized to an *N*-oxide.

### 2.4. Larvicidal Mechanism

The modes of action of insecticides are diverse; among them, the inhibitions of acetylcholinesterase (AChE), glutathione-S-transferase (GST), and nonspecific esterase activity in mosquitoes are promising insecticide mechanisms. They are important enzymes in the nervous system and are the targets for many insecticides [11,51]. The inhibition rates of **MA** and its derivatives (**4e** and **4m**) on acetylcholinesterase, glutathione-S-transferase, and nonspecific esterase activity at different concentrations were tested. As shown in Figure 5, the AChE inhibition rates of **4e** and **4m** were higher than those of **MA** at the concentrations of 250, 125, 100, and 50 μg/mL. Intriguingly, the inhibitory activities of **MA**, **4e**, and **4m** against larval enzyme AChE were concentration dependent. The inhibition rates of **MA**, **4e**, and **4m** on GST, as shown in Figure 5, were all less than 5% at the concentrations of 250, 125, 100, and 50 μg/mL and did not show a dose-dependent relationship. Similarly, the inhibition rates of **MA**, **4e**, and **4m** on nonspecific esterase were low as depicted in Figure 5. Compounds **4e** and **4m** exhibited good inhibitory activities on AChE with inhibitory rates of 59.12% and 54.30%, respectively, at the concentration of 250 μg/mL, whereas the inhibitory rate of **MA** was 9.88%. In summary, the results of the inhibition rate tests suggested that the insecticidal mechanism of **MA**, **4e**, and **4m** could be partially mediated through AChE inhibition. AChE is an important enzyme in the nervous system, hydrolyzing acetylcholine neurotransmitters and terminating nerve impulses; it is the target for both organophosphates and carbamate insecticides [52]. Insect poisoning or even death can result from the cholinergic system being destroyed or obstructed with overstimulated larval neurons, which leads to increased levels of acetylcholine in the body of the larvae as a result of decreased enzyme function. Further studies are required to validate this hypothesis.

## 3. Materials and Methods

### 3.1. Instruments and Materials

All chemical reagents were purchased from commercial supplies and utilized without further purification. **MA** was purchased from Aladdin Reagent (Shanghai, China) Co., Ltd. All reactions were monitored by thin-layer chromatography (TLC; Qingdao Haiyang Chemical, Qingdao, China), and spots were observed with UV light. Column chromatography was carried out on silica gel (200–300 or 300–400 mesh). A Bruker DPX-500 MHz instrument (Rheinstetten, German) was used to record the ^1^H NMR and ^13^C NMR spectra. HRMS spectra were measured on a Bruker micro TOF-Q instrument in electrospray ionization mode (Brooke, Switzerland). The melting point was determined using an XT-4 digital mp apparatus. *Ae. albopictus* individuals were kept in the laboratory of the International Healthcare Innovation Institute, Jiangmen, China. The larvae were fed daily with fish food. The adults were placed in a rearing cage (30 × 30 × 30 cm^3^) and received a 5% glucose solution. The mosquitoes were reared under a 14:10 light/dark photoperiod and 70% ± 5% relative humidity at 26 ± 2 °C. The female mosquito larvae of the 4th instar were used for the bioassay.

### 3.2. General Procedure for the Synthesis of MA Derivatives

#### 3.2.1. General Procedure for the Synthesis of **2a** and **2b**

**MA** (9.9348 g, 40 mmol) was added to HCl solution (6 N, 100 mL) in a 250 mL round bottom flask equipped with a stirring bar, and the stirring solution was refluxed for 6 h. TLC was used to monitor the reaction. Then, the reaction solution was decompressed and dried to remove as much water as possible. Afterward, 100 mL of methanol was added to dissolve the mixture completely, and the solution was refluxed for 4 h. The solvent was then evaporated under reduced pressure and dried under a vacuum pump for an additional 1 h. Finally, 4-chlorobenzenesulfonyl chloride or 4-bromobenzenesulfonyl chloride (60 mmol) and KOH (80 mmol) were added to the flask, and then the flask was evacuated and backfilled with nitrogen three times. Subsequently, an appropriate amount of dichloromethane (DCM) was added via a syringe. The reaction mixture was stirred overnight at room temperature. An equal amount of deionized water was added for extraction with ethyl acetate (EtOAc). The organic phase was dried with anhydrous MgSO_4_ and removed under vacuum to obtain the residue followed by purification using silica gel column chromatography (elution agent was methanol:EtOAc = 1:1) to produce the corresponding derivatives **2a** and **2b**.

Data for **2a** (C_22_H_31_ClN_2_O_4_S): yield: 36%; light brown powder; mp: 133.0–134.7 °C; ^1^H NMR (500 MHz, Chloroform-*d*) *δ*_H_ 7.87–7.75 (m, 2H), 7.50–7.42 (m, 2H), 3.67 (s, 3H), 3.64–3.57 (m, 1H), 3.53 (dd, *J* = 12.5, 5.8 Hz, 1H), 3.26 (dd, *J* = 12.5, 10.9 Hz, 1H), 2.72–2.55 (m, 3H), 2.39–2.17 (m, 2H), 2.07–2.04 (m, 1H), 2.03–1.97 (m, 1H), 1.90–1.84 (m, 2H), 1.84–1.78 (m, 2H), 1.77–1.65 (m, 1H), 1.65–1.53 (m, 2H), 1.53–1.41 (m, 2H), 1.41–1.25 (m, 5H); ^13^C NMR (126 MHz, Chloroform-*d*) *δ*_C_ 173.91, 139.06, 138.54, 128.96, 128.89, 62.97, 57.61, 56.68, 51.52, 47.44, 39.42, 34.60, 33.88, 30.83, 28.09, 27.89, 20.99, 20.80, 20.75. HRMS (ESI): C_22_H_32_ClN_2_O_4_S (455.1766) [M+H]^+^ = 455.1765.

Data for **2b** (C_22_H_31_BrN_2_O_4_S): yield: 34%; white powder; mp: 136.8–138.5 °C; ^1^H NMR (500 MHz, Chloroform-*d*) *δ*_H_ 7.77–7.71 (m, 2H), 7.66–7.60 (m, 2H), 3.67 (s, 3H), 3.63–3.56 (m, 1H), 3.53 (dd, *J* = 12.5, 5.8 Hz, 1H), 3.26 (dd, *J* = 12.5, 10.9 Hz, 1H), 2.68–2.56 (m, 3H), 2.36–2.26 (m, 1H), 2.26–2.17 (m, 1H), 2.05 (t, *J* = 3.2 Hz, 1H), 2.03–1.95 (m, 1H), 1.90–1.65 (m, 6H), 1.64–1.31 (m, 8H); ^13^C NMR (126 MHz, Chloroform-*d*) *δ*_C_ 173.91, 139.59, 131.87, 129.07, 127.00, 62.96, 57.61, 56.68, 51.54, 47.44, 39.43, 34.60, 33.88, 30.84, 28.09, 27.89, 20.99, 20.80, 20.75. HRMS (ESI): C_22_H_32_BrN_2_O_4_S (499.1261) [M+H]^+^ = 499.1265.

#### 3.2.2. General Procedure for the Synthesis of **3a** and **3b**

Compound **2a** or **2b** (10 mmol) was added to a saturated solution of NaOH in MeOH (100 mL), and the reaction solution was refluxed for 2 h until the TLC analysis showed the completion of the reaction. After the solution was cooled to room temperature, the pH value of the solution was adjusted to 7 by diluting sulfuric acid. The mixture was extracted with EtOAc and washed successively with water and brine. The organic layer was evaporated under a vacuum, and the residue was purified by flash chromatography (elution agent was methanol:EtOAc = 2:1) on silica gel to obtain the desired products **3a** and **3b**.

Data for **3a** (C_21_H_29_ClN_2_O_4_S): yield: 99%; white powder; mp: 131.4–133.2 °C; ^1^H NMR (500 MHz, Chloroform-*d*) *δ*_H_ 7.79 (d, *J* = 8.2 Hz, 2H), 7.47 (d, *J* = 8.1 Hz, 2H), 3.81–3.73 (m, 1H), 3.68–3.60 (m, 1H), 3.43–3.34 (m, 1H), 3.05 (d, *J* = 11.2 Hz, 2H), 2.47–2.43 (m, 1H), 2.22–2.10 (m, 5H), 2.06–1.98 (m, 2H), 1.94–1.81 (m, 2H), 1.77–1.69 (m, 1H), 1.72–1.61 (m, 2H), 1.61–1.53 (m, 1H), 1.49–1.33 (m, 6H); ^13^C NMR (126 MHz, Chloroform-*d*) *δ*_C_ 179.83, 138.94, 138.42, 129.16, 128.79, 62.97, 57.35, 56.28, 53.47, 46.68, 39.36, 36.13, 34.17, 31.65, 28.10, 27.72, 21.93, 20.56, 20.38. HRMS (ESI): C_21_H_30_ClN_2_O_4_S (441.1409) [M + H]^+^ = 441.1609.

Data for **3b** (C_21_H_29_BrN_2_O_4_S): yield: 99%; brown powder; mp: 134.9–136.7 °C; ^1^H NMR (500 MHz, Chloroform-*d*) *δ*_H_ 7.70 (d, *J* = 8.2 Hz, 2H), 7.63 (d, *J* = 8.3 Hz, 2H), 3.84–3.76 (m, 1H), 3.71–3.62 (m, 1H), 3.42 (t, *J* = 12.7 Hz, 1H), 3.16–3.09 (m, 2H), 2.55–2.50 (m, 1H), 2.25–2.15 (m, 2H), 2.14–2.02 (m, 4H), 1.99–1.81 (m, 3H), 1.76–1.64 (m, 3H), 1.62–1.51 (m, 1H), 1.51–1.42 (m, 2H), 1.44–1.32 (m, 3H); ^13^C NMR (126 MHz, Chloroform-*d*) *δ*_C_ 177.75, 141.09, 132.00, 128.62, 127.02, 64.29, 58.52, 55.97, 55.94, 53.47, 48.92, 39.42, 35.55, 34.95, 28.83, 27.06, 22.74, 20.04, 19.85. HRMS (ESI): C_21_H_29_BrN_2_O_4_S (485.1104) [M + H]^+^ = 485.1104.

#### 3.2.3. General Procedure for the Synthesis of **4a**–**4p**

Compound **3a** or **3b** (0.48 mmol) was reacted with different heterocyclic amines (0.60 mmol) in the presence of 1-ethyl-3-(3-dimethylaminopropyl) carbodiimide (0.60 mmol) and *N*-hydroxybenzotriazole (0.60 mmol) under nitrogen protection at room temperature, and DCM was added as the solvent. TLC was used to monitor the reaction. Then, the saturated NaHCO_3_ solution was added to the reaction mixture and extracted by EtOAc three times. The organic layer was dried with anhydrous Mg_2_SO_4_, concentrated in vacuo, and purified by column chromatography over silica gel eluted with elution agent methanol/EtOAc (*v*/*v* = 2:1) to afford the target compounds **4a**–**4p**. Data for **4a** and **4b** are presented here, whereas those for **4c**–**4p** are characterized in the Appendix A.

Data for **4a** (C_25_H_36_ClN_3_O_3_S): yield: 73%; white powder; mp: 154.4–156.7 °C; ^1^H NMR (500 MHz, Chloroform-*d*) *δ*_H_ 7.81 (d, *J* = 8.5 Hz, 2H), 7.47 (d, *J* = 8.3 Hz, 2H), 3.57–3.48 (m, 2H), 3.46 (t, *J* = 6.9 Hz, 2H), 3.44–3.35 (m, 2H), 3.19 (t, *J* = 11.5 Hz, 1H), 2.63 (d, *J* = 11.7 Hz, 1H), 2.58 (d, *J* = 10.9 Hz, 1H), 2.33–2.13 (m, 2H), 2.02 (t, *J* = 3.1 Hz, 1H), 2.00–1.84 (m, 7H), 1.86–1.82 (m, 2H), 1.84–1.74 (m, 2H), 1.71–1.60 (m, 1H), 1.52–1.38 (m, 2H), 1.41–1.25 (m, 6H); ^13^C NMR (126 MHz, Chloroform-*d*) *δ*_C_ 171.39, 138.53, 138.49, 129.11, 128.89, 62.84, 57.40, 56.70, 56.66, 47.23, 46.60, 45.59, 39.18, 34.69, 34.39, 31.40, 28.09, 27.94, 26.15, 24.43, 20.88, 20.77, 20.44. HRMS (ESI): C_25_H_37_ClN_3_O_3_S (494.2239) [M + H]^+^ = 494.2243.

Data for **4b** (C_25_H_36_ClN_3_O_4_S): yield: 55%; white powder; mp: 146.3–148.2 °C; ^1^H NMR (500 MHz, Chloroform-*d*) *δ*_H_ 7.80 (d, *J* = 8.67 Hz, 2H), 7.48 (d, *J* = 8.72 Hz, 2H), 3.73–3.58 (m, 5H), 3.61–3.44 (m, 4H), 3.17 (t, *J* = 11.7 Hz, 1H), 2.67 (d, *J* = 11.3 Hz, 1H), 2.61 (d, *J* = 11.5 Hz, 1H), 2.43–2.34 (m, 1H), 2.29–2.20 (m, 1H), 2.08–2.03 (m, 1H), 2.03–1.91 (m, 2H), 1.87–1.83 (m, 4H), 1.85–1.76 (m, 2H), 1.73–1.62 (m, 1H), 1.57–1.41 (m, 2H), 1.44–1.33 (m, 3H), 1.36–1.32 (m, 2H), 1.29 (d, *J* = 15.4 Hz, 1H); ^13^C NMR (126 MHz, Chloroform-*d*) *δ*_C_ 171.65, 138.70, 138.11, 129.12, 129.01, 66.92, 66.76, 62.90, 57.27, 56.67, 56.62, 47.52, 46.05, 41.89, 39.17, 34.45, 33.25, 31.01, 27.87, 20.83, 20.74, 20.68. HRMS (ESI): C_25_H_37_ClN_3_O_4_S (510.2188) [M + H]^+^ = 510.2192.

#### 3.2.4. General Procedure for the Synthesis of **2c** and **2d**

A solution of **2a** or **2b** (6.80 mmol) was completely dissolved with moderate DCM in a round bottom flask. Then, K_2_CO_3_ (20.40 mmol) and m-CPBA (13.6 mmol) were added and stirred for 5 min in an ice bath. The reaction system was gradually returned to room temperature and stirred overnight. TLC was applied to monitor the reaction. Then, the mixture was filtered by suction to remove excess K_2_CO_3_ and m-CPBA to obtain a crude product, which was purified by silica gel column chromatography with methanol/EtOAc (*v*/*v* = 2:1) to obtain compounds **2c** and **2d**.

Data for **2c** (C_22_H_31_ClN_2_O_5_S): yield: 90%; white powder; mp: 180.2–182.1 °C; ^1^H NMR (500 MHz, Chloroform-*d*) *δ*_H_ 7.79–7.73 (m, 2H), 7.51–7.44 (m, 2H), 5.17–5.09 (m, 1H), 4.61 (t, *J* = 12.1 Hz, 1H), 3.64 (s, 3H), 3.15–3.01 (m, 5H), 2.75 (s, 3H), 2.74–2.61 (m, 1H), 2.58–2.44 (m, 1H), 2.33–2.12 (m, 3H), 2.12–2.04 (m, 1H), 1.96–1.84 (m, 1H), 1.84–1.64 (m, 4H), 1.58–1.38 (m, 3H); ^13^C NMR (126 MHz, Chloroform-*d*) *δ*_C_ 173.91, 139.56, 138.68, 129.22, 128.41, 69.59, 69.20, 67.17, 57.10, 51.50, 50.33, 39.26, 35.38, 33.75, 29.04, 25.96, 25.31, 20.64, 17.15, 17.12. HRMS (ESI): C_22_H_32_ClN_2_O_5_S (471.1715) [M + H]^+^ = 471.1710.

Data for **2d** (C_22_H_31_BrN_2_O_5_S): yield: 90%; white powder; mp: 174.8–176.1 °C; ^1^H NMR (500 MHz, Chloroform-*d*) *δ*_H_ 7.71–7.65 (m, 2H), 7.65–7.60 (m, 2H), 5.17–5.11 (m, 1H), 4.62 (t, *J* = 12.1 Hz, 1H), 3.63 (s, 3H), 3.66–3.59 (m, 1H), 3.09 (s, 4H), 3.05 (d, *J* = 12.1 Hz, 1H), 2.74–2.62 (m, 1H), 2.57–2.44 (m, 1H), 2.52–2.48 (m, 1H), 2.32–2.10 (m, 3H), 2.10–2.03 (m, 1H), 1.95–1.83 (m, 1H), 1.81–1.63 (m, 4H), 1.57–1.37 (m, 4H); ^13^C NMR (126 MHz, Chloroform-*d*) *δ*_C_ 173.91, 140.13, 132.21, 128.51, 127.16, 69.75, 69.35, 67.09, 57.11, 51.51, 50.42, 39.31, 35.46, 33.76, 28.99, 26.02, 25.35, 20.60, 17.17, 17.14. HRMS (ESI): C_22_H_32_BrN_2_O_5_S (515.1210) [M + H]^+^ = 515.1204.

#### 3.2.5. General Procedure for the Synthesis of **3c** and **3d**

A suspension of **2c** or **2d** (5.31 mmol) in MeOH/H_2_O (80 mL) was added with NaOH (53.00 mmol), and the reaction mixture was refluxed at 110 °C and stirred for 2 h. After the TLC analysis showed the completion of the reaction, excess methanol was removed, and the pH was adjusted to 7 by HCl addition. Then, the solution was extracted with EtOAc. The organic extracts were dried and concentrated under reduced pressure. The crude products were purified by silica gel chromatography to afford **3c** and **3d** as white solids.

Data for **3c** (C_21_H_29_ClN_2_O_5_S): yield: 98%; white powder; mp: 182.3–184.4 °C; ^1^H NMR (500 MHz, Chloroform-*d*) *δ*_H_ 7.79–7.73 (m, 2H), 7.51–7.44 (m, 2H), 5.17–5.09 (m, 1H), 4.61 (t, *J* = 12.1 Hz, 1H), 3.64 (s, 3H), 3.15–3.01 (m, 4H), 2.75 (s, 2H), 2.74–2.61 (m, 1H), 2.58–2.44 (m, 1H), 2.33–2.12 (m, 3H), 2.12–2.04 (m, 1H), 1.96–1.84 (m, 1H), 1.84–1.64 (m, 4H), 1.58–1.38 (m, 3H); ^13^C NMR (126 MHz, Methanol-*d*_4_) *δ*_C_ 181.51, 138.64, 138.44, 129.03, 128.80, 68.39, 67.93, 66.16, 56.95, 54.20, 49.80, 38.69, 37.74, 34.62, 29.81, 25.30, 24.46, 21.16, 16.87. HRMS (ESI): C_21_H_30_ClN_2_O_5_S (457.1558) [M + H]^+^ = 457.1553.

Data for **3d** (C_21_H_29_BrN_2_O_5_S): yield: 97%; white powder; mp: 173.5–175.3 °C; ^1^H NMR (500 MHz, Chloroform-*d*) *δ*_H_ 7.70 (d, *J* = 8.6 Hz, 2H), 7.64 (d, *J* = 8.2 Hz, 2H), 5.15 (s, 1H), 4.10–4.07 (m, 7H), 3.11–3.09 (m, 4H), 2.46–2.42 (m, 1H), 2.31–2.28 (m, 1H), 2.25–2.21 (m, 2H), 2.04–2.00 (m, 3H), 1.49–1.43 (m, 6H); ^13^C NMR (126 MHz, Dimethyl sulfoxide-*d*_6_) *δ*_C_ 177.04, 140.43, 132.67, 129.22, 126.65, 68.47, 68.03, 65.54, 57.25, 50.23, 39.19, 37.66, 35.00, 29.90, 25.71, 24.81, 21.88, 17.22, 17.14. HRMS (ESI): C_21_H_30_BrN_2_O_5_S (501.1053) [M + H]^+^ = 501.1049.

#### 3.2.6. General Procedure for the Synthesis of **5a**–**5p**

The title compounds (**5a**–**5p**) were synthesized from intermediates **3c** and **3d** and different heterocyclic amines according to the procedure used to prepare compounds **4a**–**4p**. Data for **5a** and **5b** are presented here, whereas those for **5c**–**5p** are characterized in the Appendix A.

Data for **5a** (C_25_H_36_ClN_3_O_4_S): yield: 99%; white powder; mp: 189.2–190.7 °C; ^1^H NMR (500 MHz, Chloroform-*d*) *δ*_H_ 7.80–7.74 (m, 2H), 7.51–7.45 (m, 2H), 4.92–4.85 (m, 1H), 4.55 (t, *J* = 11.9 Hz, 1H), 3.64 (dd, *J* = 11.2, 5.1 Hz, 1H), 3.48–3.34 (m, 4H), 3.13 (t, *J* = 7.9 Hz, 2H), 3.10–3.00 (m, 3H), 2.80 (s, 2H), 2.72–2.59 (m, 1H), 2.54–2.41 (m, 1H), 2.33–2.24 (m, 1H), 2.24–2.17 (m, 1H), 2.20–2.08 (m, 3H), 2.06–1.95 (m, 1H), 1.98–1.90 (m, 2H), 1.89–1.79 (m, 2H), 1.82–1.65 (m, 2H), 1.62–1.45 (m, 4H); ^13^C NMR (126 MHz, Chloroform-*d*) *δ*_C_ 171.49, 138.74, 138.61, 129.25, 128.63, 69.57, 69.00, 67.09, 56.76, 50.06, 46.61, 45.60, 38.78, 35.00, 34.51, 29.47, 26.13, 26.05, 25.25, 24.43, 19.62, 17.24, 17.17. HRMS (ESI): C_25_H_37_ClN_3_O_4_S (510.2188) [M + H]^+^ = 510.2186.

Data for **5b** (C_25_H_36_ClN_3_O_5_S): yield: 84%; white powder; mp: 178.6–180.4 °C; ^1^H NMR (500 MHz, Chloroform-*d*) *δ*_H_ 7.79–7.72 (m, 2H), 7.52–7.46 (m, 2H), 4.76–4.69 (m, 1H), 4.39 (t, *J* = 11.9 Hz, 1H), 3.72–3.59 (m, 4H), 3.62–3.56 (m, 1H), 3.56–3.41 (m, 2H), 3.34–3.21 (m, 2H), 3.09 (d, *J* = 10.7 Hz, 3H), 2.65–2.55 (m, 1H), 2.50–2.38 (m, 1H), 2.38–2.27 (m, 1H), 2.26–2.18 (m, 1H), 2.21–2.13 (m, 1H), 2.15–2.07 (m, 2H), 2.06–1.95 (m, 1H), 1.93 (s, 1H), 1.84–1.73 (m, 2H), 1.75–1.69 (m, 2H), 1.65–1.48 (m, 3H), 1.35–1.18 (m, 2H); ^13^C NMR (126 MHz, Chloroform-*d*) *δ*_C_ 171.57, 138.91, 138.21, 129.32, 128.62, 68.92, 68.41, 67.33, 66.88, 66.73, 56.58, 49.84, 45.97, 41.86, 38.53, 34.65, 32.98, 29.57, 25.86, 25.10, 19.88, 17.15, 17.08. HRMS (ESI): C_25_H_37_ClN_3_O_5_S (526.2137) [M + H]^+^ = 526.2131.

#### 3.2.7. General Procedure for the Synthesis of **6a**–**6p**

Anhydrous DCM (5 mL) was added to a 100 mL two-outlet flask with **5a**–**5p** (0.59 mmol) under nitrogen protection. Each compound was completely dissolved, and the solution was stirred for 5 min in a cold bath. Then, TFAA (1.17 mmol) was injected, and the solution was subjected to an ice bath for another 3.5 h. The solvent was drained by a vacuum pump for 1 h after the reaction and then sealed with nitrogen gas. Anhydrous DCM (5 mL) was added, and the solution was stirred for 5 min in an ice bath. Then, Et_3_N (0.06 mmol) and TMSCN (1.77 mmol) were added sequentially. TLC was utilized to monitor the reaction. Saturated NaHCO_3_ (15 mL) was added for the quenching reaction. The product was extracted with EtOAc, dried with anhydrous Mg_2_SO_4_, filtered by a sand core funnel, and purified by column chromatography (methanol:EtOAc = 1:8) to collect compounds **6a**–**6p**. Data for **6a** and **6b** are presented here, whereas those for **6c**–**6p** are characterized in the Appendix A.

Data for **6a** (C_26_H_35_ClN_4_O_3_S): yield: 69%; white powder; mp: 155.3–157.6 °C; ^1^H NMR (500 MHz, Chloroform-*d*) *δ*_H_ 7.84–7.78 (m, 2H), 7.53–7.47 (m, 2H), 4.06–3.99 (m, 1H), 3.64–3.57 (m, 1H), 3.46 (t, *J* = 6.9 Hz, 3H), 3.44–3.36 (m, 1H), 3.38–3.31 (m, 1H), 3.10 (dd, *J* = 15.2, 12.2 Hz, 1H), 2.68–2.60 (m, 2H), 2.43–2.28 (m, 2H), 2.28–2.16 (m, 2H), 2.04–1.90 (m, 3H), 1.90–1.81 (m, 4H), 1.76–1.39 (m, 8H), 1.38–1.25 (m, 2H); ^13^C NMR (126 MHz, Chloroform-*d*) *δ*_C_ 170.88, 139.26, 139.17, 129.53, 128.65, 116.33, 64.62, 57.34, 51.61, 50.95, 46.59, 45.62, 45.58, 43.13, 42.74, 33.89, 26.53, 26.12, 25.16, 24.53, 24.41, 24.12, 23.90, 22.76. HRMS (ESI): C_26_H_36_ClN_4_O_3_S (519.2191) [M + H]^+^ = 519.2184.

Data for **6b** (C_26_H_35_ClN_4_O_4_S): yield: 80%; white powder; mp: 150.5–152.7 °C; ^1^H NMR (500 MHz, Chloroform-*d*) *δ*_H_ 7.84–7.77 (m, 2H), 7.54–7.48 (m, 2H), 4.07–3.99 (m, 1H), 3.71–3.64 (m, 5H), 3.66–3.51 (m, 3H), 3.51–3.38 (m, 2H), 3.10 (dd, *J* = 15.2, 12.2 Hz, 1H), 2.68–2.59 (m, 2H), 2.43–2.21 (m, 5H), 2.08–1.94 (m, 1H), 1.84–1.77 (m, 1H), 1.76–1.68 (m, 2H), 1.72–1.65 (m, 1H), 1.68–1.53 (m, 4H), 1.54–1.43 (m, 1H), 1.40–1.24 (m, 2H); ^13^C NMR (126 MHz, Chloroform-*d*) *δ*_C_ 171.06, 139.26, 129.60, 128.57, 116.29, 66.92, 66.67, 64.55, 57.22, 51.59, 50.94, 45.96, 45.47, 43.14, 42.58, 41.91, 32.19, 26.52, 25.16, 24.51, 24.11, 23.87, 22.90. HRMS (ESI): C_26_H_36_ClN_4_O_4_S (535.2140) [M + H]^+^ = 535.2133.

### 3.3. Bioassay

#### 3.3.1. Insecticidal Tests for Larvae of *Ae. albopictus*

The larvicidal activity of **MA** and its derivatives against the 4th instar larvae was evaluated using established techniques with minor modifications [53,54,55]. A 24-well plate with a test well was used. Four replication wells were allotted for each derivative, and each well had five larvae. Then, 985 μL of clean deionized water, 5 μL of feed solution (25 mg/mL), and 10 μL of derivative solution were added. Deltamethrin and dimethylsulfoxide replaced the derivative as negative and positive control groups, respectively. Three independent replicate tests were carried out. The 24-well plate was cultivated in an incubator maintained at the constant temperature of 28 °C and 80% relative humidity under 12 h light and 12 h dark. After 24 h, the lethality of each derivative for the larvae was recorded.

After the pre-experiment screening, **MA** and serval derivatives were chosen to participate in the LC_50_ test. First, stock solutions with a range of concentrations were created by dissolving **MA** and its derivatives in dimethylsulfoxide (100, 50, 25, and 12.5 mg/mL, respectively). Second, 1 mL of each stock solution was added to 99 mL of distilled water to create the test solutions. Third, 20 4th instar larvae were inserted into each test solution, and triplicate mortality checks were carried out after 24 h of incubation. Eight to eleven concentrations of each chemical were tested.

#### 3.3.2. Insecticidal Tests for Adult *Ae. albopictus*

The activities of **MA** and its derivatives against adult mosquitoes were evaluated using the bottle bioassay following the stated techniques with minor modifications [53,54,55]. **MA** and its derivatives were separately dispersed in dimethylsulfoxide to create stock solutions (100, 50, 25, and 12.5 mg/mL, respectively). Second, a 250 mL Wheaton bottle was filled with 1 mL of each stock solution. A consistent thin coating formed on the inner surface of the container after the solvent was volatilized for 1 h at room temperature while shaking and rotating the bottle. Third, each bottle was exposed to 20 non-blood-fed female mosquitoes (2–5 days old) for 2 h. The insects were then moved to culture cups and raised in the incubator. The mortality was recorded after 24 h of rearing at 26–28°C, 80% relative humidity, and light:dark (12 h:12 h). Deltamethrin and dimethylsulfoxide were used as negative and positive control groups, respectively. Importantly, the mortality rate of the negative control group should not exceed 5%. Three sets of repeated tests were completed for different batches of adult mosquitoes.

#### 3.3.3. Effects of Partial MA Derivatives on the Growth Cycle of *Ae. albopictus*

Effects on the Emergence of *Ae. albopictus* Larvae

Compounds **4e** and **4m** were filtered out to study the impacts on the emergence of *Ae. albopictus* larvae because they had stronger larvicidal action than the other **MA** derivatives. The high-throughput screening method [56,57] with the outcome of the LC_50_ test was used to determine the final test concentration of the derivative, which was set at LC_30_. Five *Ae. albopictus* larvae in the 4th instar were reared for 24 h in an incubator with constant temperature and humidity. Then, 985 μL of deionized water, 10 μL of sample solution, and 5 μL of feed solution were added to each well in the 24-well plate. For each concentration, eight replicate wells were set up, and three separate replicate experiments were run.

The still alive larvae were removed with a dropper and cleaned 2–3 times in deionized water after being cultured for 24 h. Then, the larvae were moved to a fresh 24-well plate, and a treated larva was placed in each well along with 1900 μL of deionized water and 10 μL of feed solution. The identically treated 24-well plate of larvae was placed in each mosquito cage at the same time, along with 10% sugar water. The temperature, relative humidity, and length of light and dark periods in the rearing environment were fixed at 28 °C, 80%, and 12 h each, respectively. Larval status was scored as follows: 0: death, 4: larva, 5: pupa, 6: adult mosquito, 4-0: death as larva, 5-0: death as pupa, and 6-0: death as adult mosquito.

Effects on Fecundity of Adult Female *Ae. albopictus*

The mosquitoes that evolved from the larvae that endured the aforementioned trials were starved for 24 h and then fed with blood to the point that they became visibly blood-red [58]. At this point, a manual suction apparatus was used to transport the sucked female mosquitoes to fresh cages. Each cage contained five female mosquitoes, an egg collector, and a water-feeding apparatus. The following formula was used to determine the fertility of females based on the average number of eggs laid by females:

Average number of eggs laid (%) = number of eggs on the oviposition paper/number of females laying eggs × 100%.

### 3.4. Mechanism for Killing Larvae by Test Enzymatic Activity

Acetylthiocholine iodide was used as the substrate and dithiobisnitrobenzoic acid (DTNB) was used as the chromogen to measure AChE activity according to the methods described by Ellman et al. [59]. The techniques described by Polson et al. [60] were used to measure GST activity using 1-chloro-2,4-dinitrobenzene (CDNB) as the substrate. The method established by Azratul-Hizayu et al. was used to measure nonspecific esterase activity using *α*-naphthalene acetate [61]. A microplate reader was used to perform each test in triplicate.

### 3.5. Statistical Analysis

The larvicidal and adulticidal effects for lethal bioassays were recorded 24 h after treatment. Data obtained from each dose–larvicidal bioassay were subjected to probit analysis; LC_10–50_, LC_90_ values, and slopes were generated. Data from the growth cycle of *Ae. albopictus* and enzymatic activity were obtained referring to the above sections. All analyses were conducted using the statistical package SPSS 14.0 [62]. The statistical value of *p* < 0.05 was considered as significantly different.

## 4. Conclusions

In conclusion, **MA** derivatives, including 10 intermediates and 48 target compounds in three series, were designed, synthesized, and evaluated for their anti-mosquito activities against *Ae. albopictus*. Compounds **4b**, **4e**, **4f**, **4m**, **4n**, **6e**, **6k**, **6m**, and **6o** demonstrated higher larvicidal activity against *Ae. albopictus* than the other compounds. The LC_50_ values of compounds **4m**, **4e**, and **6m** reached 140.08, 147.65, and 205.79 μg/mL, respectively, whereas the LC_50_ value of **MA** was 659.34 μg/mL. The test on larval emergence showed that the selected **MA** derivatives delayed the emergence time and reduced the emergence rate of *Ae. albopictus* larvae. The resulting mortalities suggested that the chronic toxicity of the selected **MA** derivatives would cause the larvae to fail to transform into pupae and emerge successfully. The results of **MA**, **4e**, and **4m** in inhibiting oviposition indicated that these compounds exhibited a clear effect on the fecundity of female *Ae. Albopictus*. A dose-dependent relationship was observed between the compounds and the oviposition of *Ae. albopictus*. However, our findings indicated that more research on **MA** derivatives against adult mosquitoes is required.

The SAR analysis showed that the introduction of unsaturated heteroatom rings into the carboxyl group after D ring opening could enhance larvicidal activity. However, the **MA** derivatives whose N-1 was oxidized lost their anti-mosquito capabilities, suggesting that maintaining the bareness of N-1 was essential to preserve anti-mosquito activity. The addition of a cyan group at C-6 or a benzene sulfonyl group at N-16 did not substantially change anti-mosquito activity. Additionally, at the concentration of 250 μg/mL, compounds **4e** and **4m** showed good AChE inhibitory rates of 59.12% and 54.30%, respectively, whereas **MA** had an inhibitory rate of 9.88%. Therefore, this study paves the way for further structural modifications of **MA** as potential botanical anti-mosquito agents in continued study and future development.

## Data Availability

Not applicable.

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
