# Peer review of "Novel Matrine Derivatives as Potential Larvicidal Agents against Aedes albopictus: Synthesis, Biological Evaluation, and Mechanistic Analysis"

_molecules, 2023, doi:10.3390/molecules28073035_

Round 1
Reviewer 1 Report
Report
The main concern of this research article is that none of the obtained results (for Biological evaluation assays) were discussed in the results and discussion section which consider a critical issue stands behind further consideration as a scientific paper. However, I have reviewed this manuscript and my comments are as the following:
Abstract:
· Please give a background on why this particular insect was selected, you may refer to its importance in disease transmission.
· Line 18: “to develop new potential anti-mosquito agents” replace with “to develop novel mosquitocidal agents”.
· Line 19: Delete “and their anti-mosquito activities were evaluated”.
· Methods used for insecticidal activity evaluation are not provided here.
· Line 24: replace “anti-mosquito” with “mosquitocidal” here and in each subsequent place in the manuscript.
· Lines 30-31: “the structural modification and anti-mosquito activity of MA and its derivatives provided a strong foundation for the further research and development of MA and its derivatives as plant-derived anti-mosquito candidates.” You may replace this with “the structural modification and mosquitocidal activity of MA and its derivatives obtained here paves the way for those seeking strong mosquitocidal agents from plant origin.”
Introduction
· Line 56: Delete “including people”.
· Lines 62-63: “which make MA have a great potential for development into an anti-mosquito agent” replace with “which make them potential candidates as mosquitocidal agents”.
· Lines 63-65: “In the present study, we designed and synthesized three series of MA derivatives by structural modification, and the derivatives were characterized and screened for their insecticidal activity against larva and adult mosquitoes.” replace with “In the present study, we designed, synthesized and characterized three series of MA derivatives by structural modification, and screened these derivatives for their potential larvicidal and adulticidal activity against the mosquito vector, Aedes albopictus.”
Results and Discussion
· For the bioassays section, there is no discussion included.
· Line 92: “Insecticidal tests for larvae and adult mosquitos”. User either “Insecticidal activities” OR “larvicidal and adulticidal activities”
· Did you consider the effect of tested materials on the non-target organisms?
· Lines 93-114: Consider CRITICAL language revision of these paragraphs.
· Did you used a negative and positive control for larvicidal and adulticidal toxicity? negative and positive controls were absent, compromising the statistical magnitude of the results presented. Insert negative and positive controls into biological assays if you already tested them.
· Line 134: for “effects on the emergence of A. albopictus larvae”. Emergence from what? Do you mean larval development/survival? Please consider revision. Where is the positive control group for this test?
· Line 153: “fertility” is not appropriate synonym, you mean fecundity (Number of eggs laid per female). Consider revision.
Tables
· Identify all abbreviations inserted in table 1 and table 2 such as, R1, R2, SD, LC20 and other abbs.
Materials and Methods
· Line 403: Please mention the original references for the larvicidal toxicity evaluation bioassays and then cite yours as a modification on it. You can see the WHO report published in 2005, and the same for the adulticidal toxicity. World Health Organization. Guidelines for Laboratory and Field Testing of Mosquito Larvicides; World Health Organization: Geneva, Switzerland. 2005, pp. 1–39.
· What was the original concentrations used to evaluate the lethal concentrations used later (LCs)? Name it in the methods section.
· Sample size is different for each experiment, you may add a separate column indicating the S.S. for each treatment in tables.
· Still, positive control was not tested.

Author Response
Dear reviewer,
Many thanks for your comments concerning our manuscript entitled “Novel matrine derivatives as potential larvicidal agents against Aedes albopictus: Synthesis, biological evaluation, and mechanistic analysis”. The comments are valuable and constructive for improving our manuscript. We have looked into your comments and suggestions carefully and made some changes accordingly. Revised parts are marked in red in the 'Revised Manuscript with Track Changes' file and the Supplementary material file. Response to address the concerns of the reviewers are outlined below:
Abstract:
Comment 1: Please give a background on why this particular insect was selected, you may refer to its importance in disease transmission.
Response: Thank you for your valuable suggestion. We have added the background of the Aedes albopictus according to its importance in disease transmission. Please see the lines 19-21 in “Revised Manuscript with Track Changes” file.
Comment 2: Line 18: “to develop new potential anti-mosquito agents” replace with “to develop novel mosquitocidal agents”.
Response: We have made revision as your suggestion. Please see the line 18 in “Revised Manuscript with Track Changes” file.
Comment 3: Line 19: Delete “and their anti-mosquito activities were evaluated”.
Response: We have deleted it. Please see the line 19 in “Revised Manuscript with Track Changes” file.
Comment 4: Methods used for insecticidal activity evaluation are not provided here.
Response: We have added the methods used for insecticidal activity evaluation. Please see the line 21 in “Revised Manuscript with Track Changes” file.
Comment 5: Line 24: replace “anti-mosquito” with “mosquitocidal” here and in each subsequent place in the manuscript.
Response: We have made revision as your suggestion. Please see the line 25 and other subsequent place in the manuscript in “Revised Manuscript with Track Changes” file.
Comment 6: Lines 30-31: “the structural modification and anti-mosquito activity of MA and its derivatives provided a strong foundation for the further research and development of MA and its derivatives as plant-derived anti-mosquito candidates.” You may replace this with “the structural modification and mosquitocidal activity of MA and its derivatives obtained here paves the way for those seeking strong mosquitocidal agents from plant origin.”
Response: We have made revision as your suggestion. Please see the lines 30-32 in the manuscript in “Revised Manuscript with Track Changes” file.
Introduction:
Comment 7: Line 56: Delete “including people”.
Response: Thank you for your suggestion. We have deleted it in “Revised Manuscript with Track Changes” file.
Comment 8: Lines 62-63: “which make MA have a great potential for development into an antimosquito agent” replace with “which make them potential candidates as mosquitocidal agents”.
Response: We have made revision as your suggestion. Please see the lines 64-65 in the manuscript in “Revised Manuscript with Track Changes” file.
Comment 9: Lines 63-65: “In the present study, we designed and synthesized three series of MA derivatives by structural modification, and the derivatives were characterized and screened for their insecticidal activity against larva and adult mosquitoes.” replace with “In the present study, we designed, synthesized and characterized three series of MA derivatives by structural modification, and screened these derivatives for their potential larvicidal and adulticidal activity against the mosquito vector, Aedes albopictus.”
Response: We have made revision as your suggestion. Please see the lines 65-68 in the manuscript in “Revised Manuscript with Track Changes” file.
Results and Discussion:
Comment 10: For the bioassays section, there is no discussion included.
Response: Thank you for your comments. We have added more discussions in every subheading of the bioassays section.
Comment 11: Line 92: “Insecticidal tests for larvae and adult mosquitos”. User either “Insecticidal activities” OR “larvicidal and adulticidal activities”
Response: We have made revision as your suggestion. Please see the line 94 in the manuscript in “Revised Manuscript with Track Changes” file.
Comment 12: Did you consider the effect of tested materials on the non-target organisms?
Response: Thank you for your comments. MA was considered as a promising natural product with various pharmacological activities [1] and we will investigate the effect of tested MA derivatives on the non-target organisms in our further research, including aquatic organism. Acute toxicity test on the gill tissues of Danio rerio will be studied [2].
[1] Zhang H, Chen L, Sun X, et al. Matrine: a promising natural product with various pharmacological activities [J]. Frontiers in pharmacology, 2020, 11: 588.
[2] Hazarika H, Krishnatreyya H, Tyagi V, et al. The fabrication and assessment of mosquito repellent cream for outdoor protection [J]. Scientific Reports, 2022, 12(1): 2180.
Comment 13: Lines 93-114: Consider CRITICAL language revision of these paragraphs.
Response: We have made revision as your suggestion. Please see the Lines 95-122 in “Revised Manuscript with Track Changes” file.
Comment 14: Did you used a negative and positive control for larvicidal and adulticidal toxicity? negative and positive controls were absent, compromising the statistical magnitude of the results presented. Insert negative and positive controls into biological assays if you already tested them.
Response: Negative and positive controls study were also performed during our biological assays; these results were also introduced. Please see the Table 1 in “Revised Manuscript with Track Changes” file.
Comment 15: Line 134: for “effects on the emergence of Ae. albopictus larvae”. Emergence from what? Do you mean larval development/survival? Please consider revision. Where is the positive control group for this test?
Response: Thank you for your suggestion. “Effects on the emergence of Ae. albopictus larvae” means that how the survival situation of the larva would be in a period of time after treated by compounds at concentration of LC30. We have made revisions in “2.2.3.1. Effects on the emergence of Ae. albopictus larva” of “Revised Manuscript with Track Changes” file. The result of this experiment indicated the different effects on the emergence of Ae. albopictus larvae between MA and its derivatives and there was no appropriate drug as positive control group. The negative control group (dimethylsulfoxide) was conducted and we referred to the literature bellow:
Levi T, Ben-Dov E, Shahi P, et al. Growth and development of Aedes aegypti larvae at limiting food concentrations [J]. Acta Tropica, 2014, 133: 42-44.
Comment 16: Line 153: “fertility” is not appropriate synonym, you mean fecundity (Number of eggs laid per female). Consider revision.
Response: We have made revision as your suggestion. Please see the Lines 179 in “Revised Manuscript with Track Changes” file.
Tables:
Comment 17: Identify all abbreviations inserted in table 1 and table 2 such as, R1, R2, SD, LC20 and other abbs.
Response: We have made revision as your suggestion. Please see Table 1 and Table 2 in “Revised Manuscript with Track Changes” file.
Materials and Methods:
Comment 18: Line 403: Please mention the original references for the larvicidal toxicity evaluation bioassays and then cite yours as a modification on it. You can see the WHO report published in 2005, and the same for the adulticidal toxicity. World Health Organization. Guidelines for Laboratory and Field Testing of Mosquito Larvicides; World Health Organization: Geneva, Switzerland. 2005, pp. 1–39.
Response: We have made revision as your suggestion. Please see line 453 and the References part in “Revised Manuscript with Track Changes” file.
Comment 19: What was the original concentrations used to evaluate the lethal concentrations used later (LCs)? Name it in the methods section.
Response: We have added original concentrations in lines 464-465 and 473. Please see them in “Revised Manuscript with Track Changes” file.
Comment 20: Sample size is different for each experiment, you may add a separate column indicating the S.S. for each treatment in tables.
Response: We have added sample size in Table 2. Please see them in “Revised Manuscript with Track Changes” file.
Comment 21: Still, positive control was not tested.
Response: Thank you very much, and we have revised. Please see Materials and Methods (lines 456-458) in “Revised Manuscript with Track Changes” file.
Best regards,
Panpan Wu
Reviewer 2 Report
Authors represent a study on a structural modification and anti-mosquito activities of matrine and its derivates as plant-derivated pesticide. Manuscript is clear and relevant to the field of public pest control and chemical technology. Cited references are recent and relevant. The experimental design is appropriate and material and methods are mostly good explained. Results are clear, but some statistical values which indicate effects on fertility and larvicidal effect are missing. Also the description of the statistical analysis is missing within the section materials and methods. Suggestions and some detailed comments are listed within the attached file. After the authors make changes according to the comments and suggestions, the manuscript will be appropriate for publishing in the journal Molecules.

Author Response
Dear reviewer,
Many thanks for your comments concerning our manuscript entitled “Novel matrine derivatives as potential larvicidal agents against Aedes albopictus: Synthesis, biological evaluation, and mechanistic analysis”. The comments are valuable and constructive for improving our manuscript. We have looked into your comments and suggestions carefully and made some changes accordingly. Revised parts are marked in red in the 'Revised Manuscript with Track Changes' file and the Supplementary material file. Response to address the concerns of the reviewers are outlined below:
Comment 1: Line 36 Add “from the plant species of the family: Fabaceae” between from and Sophora. Also, indicate authority for each species when it is first mentioned; Sophora flavescens Aiton; Sophora tonkinesis Gagnep; Sophora alopecuroides L.
Response: Thank you for your detailed suggestions. We have revised. Please see the lines 37-38 in “Revised Manuscript with Track Changes” file.
Comment 2: Line 41 The same as previous, add authority for mentioned pest species.
Response: We have made revision as your suggestion. Please see the lines 43-45 in “Revised Manuscript with Track Changes” file.
Comment 3: Line 44 Change touch killing with contact toxicity
Response: We have made revision as your suggestion. Please see the line 46 in “Revised Manuscript with Track Changes” file.
Comment 4: Line 45 Change in killing insects with against insects
Response: We have made revision as your suggestion. Please see the line 48 in “Revised Manuscript with Track Changes” file.
Comment 5: Line 46 delete abbreviation A. albopictus, and add authority (Skusse)
Response: We have made revision as your suggestion. Please see the line 49 in “Revised Manuscript with Track Changes” file.
Comment 6: Line 48 Add …the mosquito… between of and family; Change Culicidae with Cullicidae
Response: We have made revision as your suggestion. Please see the line 50 in “Revised Manuscript with Track Changes” file.
Comment 7: Line 50 Write chikungunya with the first an uppercase initial letter
Response: Thank you very much, and we have revised. Please see the line 52 in “Revised Manuscript with Track Changes” file.
Comment 8: Line 128 Switch places of values as follows: 4.47, 4.71
Response: We have made revision as your suggestion. Please see the line 141 in “Revised Manuscript with Track Changes” file.
Comment 9: Results - Effects on fertility of adults: display the F, df and p values for the statement that average egg-laying rate was decreased remarkably
- Larvicidal mechanism: also F, df and p values are missing
Response: Thank you for your valuable suggestion. We have showed significant differences in parts of “Effects on fertility of adults” and “Larvicidal mechanism”. Please see them in “Revised Manuscript with Track Changes” file.
Comment 10: Line 226 Change another with additional
Response: Thank you very much, and we have revised. Please see the line 276 in “Revised Manuscript with Track Changes” file.
Comment 11: Materials and Methods: Add additional paragraph:
3.5. Statistical analysis, where you will describe in detail which program, software, and tests, did you use for the analysis of results obtained in bioassay tests….what about the significance level…..
Response: We have added additional paragraph. Please see the 3.5. Statistical analysis part in “Revised Manuscript with Track Changes” file.
Best regards,
Panpan Wu
Round 2
Reviewer 1 Report
All comments have been addressed.